# The early predictive value of frailty for health-related quality of life among elderly patients with cancer receiving curative chemotherapy

Yi-Cheng Hu[1,2◉], Shih-Ying Chen[3◉], Wen-Chi Chou[4,5], Jen-Shi Chen🄳[4,5], Li-Chueh Weng[3,6], Pei-Kwei Tsay[7,8], Woung-Ru Tang🄳[3,9]*

1 Graduate Institute of Clinical Medical Sciences, Chang Gung University, Taoyuan, Taiwan, 2 Department of Nursing, Ching Kuo Institute of Management and Health, Keelung, Taiwan, 3 School of Nursing, College of Medicine, Chang Gung University, Taoyuan, Taiwan, 4 Department of Hematology-Oncology and Cancer Center, Chang Gung Memorial Hospital Linkuo Branch, Taoyuan, Taiwan, 5 College of Medicine, Chang Gung University, Taoyuan, Taiwan, 6 Department of General Surgery, Chang Gung Memorial Hospital Linkuo Branch, Taoyuan, Taiwan, 7 Department of Public Health and Center of Biostatistics, College of Medicine, Chang Gung University, Taoyuan, Taiwan, 8 Department of Pediatrics, Chang Gung Children's Hospital, Chang Gung University, Taoyuan, Taiwan, 9 Department of Nephrology, Chang Gung Memorial Hospital Linkuo Branch, Taoyuan, Taiwan

◉ These authors contributed equally to this work.
* wtang@mail.cgu.edu.tw

**Data Availability Statement:** All relevant data are within the paper and its Supporting Information files.

## Abstract

Changes in health-related quality of life (HRQOL) among elderly patients with cancer before and after receiving curative treatment, such as chemotherapy, have always been an important consideration in physician–patient treatment decision-making. Although frailty assessment can help predict the effects of chemotherapy, there is a lack of relevant literature on its effectiveness in predicting post-chemotherapy HRQOL. Therefore, this study investigated the early predictive value of pre-chemotherapy frailty assessment for post-chemotherapy HRQOL among elderly patients with cancer receiving curative chemotherapy. From September 2016 to November 2018, this study enrolled elderly patients with cancer aged $\geq 65$ years (N = 178), who were expected to receive chemotherapy at three hospitals in Taiwan. The mean age of patients was 71.70 years (SD = 5.46 years) and half of them were female (n = 96, 53.9%). A comprehensive geriatric assessment was performed to measure frailty in 178 participants one week before receiving chemotherapy ($T_0$). Further, the HRQOL of the elderly patients with cancer was assessed again, four weeks after chemotherapy ($T_1$). After controlling for demographic variables, this study evaluated the predictive value of frailty for HRQOL using a hierarchical regression analysis. A total of 103 (57.9%) participants met the frailty criteria. The results showed that 31.1%–56.7% of the variance in the seven domains of HRQOL could be explained by demographic variables and the presence or absence of frailty. This suggests that the presence or absence of frailty is an important predictor of the illness burden domain (β = 9.5; p < .05) of HRQOL. Frailty affects the illness burden domain of HRQOL in elderly patients with cancer. Finally, the administration of frailty assessments before treatment is recommended as a reference for patient treatment decision-making.

**Funding:** The author(s) received no specific funding for this work.

**Competing interests:** The authors have declared that no competing interests exist.

## Introduction

In March 2018, the proportion of the elderly adults in Taiwan exceeded 14%, and Taiwan officially became an aging society. Taiwan is expected to become a super-aged society in 2030. Compared with Japan, which has the largest elderly population in the world, Taiwan is estimated to have transformed into a super-aged society three years earlier [1]. The increase in the elderly population has become an important issue in several countries worldwide [2]. Notably, two-thirds of the patients with cancer worldwide are older than 65 years [3], and it is estimated that the number of elderly patients with cancer will increase by 14 million by 2035 [4]. Since 1982, cancer has been the leading cause of death in Taiwan, and the median age at death has increased annually [5]. With an increase in the aging population and incidence of cancer, the importance of assessing elderly patients with cancer receiving anticancer treatment (e.g., radical resection, radiotherapy, and chemotherapy) has increased, as some patients cannot endure intensive treatment and thus experience severe life-threatening complications [6, 7].

Although the effectiveness of chemotherapy in cancer control has been demonstrated in many clinical trials [8], treatment-related side effects can reduce the health-related quality of life (HRQOL) of patients with cancer, especially the elderly [9]. The HRQOL of elderly patients with cancer is unique [3]. Therefore, the European Organization for Research and Treatment of Cancer (EORTC) developed a specific measurement tool for HRQOL in elderly patients with cancer: the EORTC QLQ-ELD14 [3, 10]. However, the relevant literature has mainly focused on cross-sectional studies [10–12], overlooking longitudinal studies.

Frailty refers to a clinical symptom in which the function and reserve capacity of multiple physiological systems are diminished, affecting the ability to maintain body stability while responding to stressful events or leading to several adverse postsurgical outcomes [13]. Scholars have emphasized that frailty is correlated with aging [14] and that the level of frailty in elderly patients with cancer is negatively correlated with HRQOL [15]. Scholars have conducted a systematic review of frailty-related studies; one study found that the mean prevalence of frailty in elderly patients with cancer was 43% [16], and another reported a rate of 68% [17]. Frailty assessment helps predict the effects of chemotherapy in patients with cancer [18]. Therefore, experts have encouraged elderly patients with cancer to undergo a frailty assessment before treatment [6, 19]. The American Society of Clinical Oncology (ASCO), International Society of Geriatric Oncology (SIOG), and National Comprehensive Cancer Network (NCCN) recommend a comprehensive geriatric assessment (CGA) as the gold standard for frailty assessment [19–21]. However, the CGA includes an extensive number of domains— functional state [19–21], social support [21], mood [19–21], comorbidity [19–21], cognition [19–21], nutrition [19–21], falls [20], and polypharmacy [21]—which leads to time-consuming assessments. Moreover, the CGA can only be performed by trained professionals, resulting in limited clinical application [22]. Considering these issues, scholars have suggested that there is no need to assess all the domains of the CGA and that assessment can only be performed on important domains [23]. A systematic review indicated that the CGA should include the assessment of at least five domains, and failure to meet the criteria of two domains should be considered the defining criteria of frailty [16].

Changes in the HRQOL of patients with cancer before and after treatment have always been an important issue for medical teams, patients, and their family members [7], and an important consideration in treatment decision-making for elderly patients with cancer [24]. Therefore, there is a need to investigate the predictive value of frailty for post-treatment HRQOL in elderly patients with cancer before chemotherapy. The results can be used as a

reference for treatment-related decision-making by clinicians for elderly patients with cancer (providing anticancer treatment, reducing anticancer treatment intensity, or not providing anticancer treatment) [6], which was the main motivation for this study.

## Methods

### Participants and data collection

This study used part of the data from a large-scale prospective, longitudinal, and observational study (frailty assessment for high-risk patients with cancer undergoing curative intent antitumor treatment) for analysis [25]. This study administered a structured questionnaire to understand the frailty of patients with cancer before treatment and the changes in HRQOL before and after treatment. Purposive sampling was used to enroll patients from three hospitals in Taiwan who met the inclusion criteria: (1) patients with solid tumor or lymphoma aged $\geq$ 65 years who were expected to receive chemotherapy, (2) patients who were willing to sign the informed consent form, and (3) patients who were conscious and able to complete the questionnaire interviews. Conversely, the following exclusion criteria were used: (1) patients with cancer with cognitive impairment who could not complete the questionnaire interviews, and (2) patients whose objective was not to cure cancer.

After the protocol was reviewed and approved by an institutional review board (no. 201600916B0), the research assistants explained the research purpose and methods to the patients. The patients completed the questionnaire after signing an informed consent form. The research assistants performed the first assessment within one week before the patients received chemotherapy ($T_0$). The assessment included a basic attribute information sheet, frailty assessment, and HRQOL scale. Four weeks after the patients received chemotherapy, another assessment ($T_1$) was conducted to collect data using the HRQOL scale.

### CGA

The CGA was used to assess seven domains: functional status (activities of daily living/instrumental activities of daily living), nutrition, comorbidity, falls, polypharmacy, social support, and mood [19, 21]. Since relevant societies do not recommend the use of consistent domains, following the suggestion of scholars that the CGA should assess at least five domains, this study chose the above-mentioned five domains with the highest proportion of CGA impairment. Further, this study deemed patients with impairments in more than two domains frail [16, 17]. The assessment tools and cut-off standards for the five domains considered in this study are listed in Table 1.

### HRQOL scale

This study assessed the HRQOL of elderly patients with cancer using the EORTC QLQ-ELD14 developed by the European Organization for Research and Treatment. The scale comprises 14 items asking patients about their HRQOL in the past week, each with a scoring range of 1–4 points, and is divided into five multi–item scales (three questions for mobility, two questions for worries about others, three questions for future worries, two questions for maintaining purpose, and two questions for illness burden) and two single items (joint stiffness and family support). The scale score was converted to 0–100 points based on the EORTC score conversion criteria. Higher scores indicated worse HRQOL (domains of mobility, worries about others, future worries, illness burden, and joint stiffness). Conversely, for the domains of maintaining purpose and family support, the higher the score, the better the HRQOL [10].

**Table 1. Measures of comprehensive geriatric assessment.**

| Frailty domain | Measure | Score range | Cut-off value |
|---|---|---|---|
| Functional status | ADL | 0–100 | < 100 |
| | IADL | 0–8 | < 8 |
| Nutrition | MNA-SF | 0–14 | < 12 |
| Comorbidity | CCI | 0–37 | > 1 |
| Falls | Number of falls | 0–∞ | > 1 |
| Polypharmacy | Number of medications | 0–∞ | > 4 |
| Social support | Living alone | Yes/No | Yes |
| Mood | GDS-4 | 0–4 | > 0 |

Note. ADL, activities of daily living; IADL, instrumental activities of daily living; MNA-SF, Mini Nutritional Assessment-Short Form; CCI, Charlson Comorbidity Index; GDS, Geriatric Depression Scale.

## Statistical analysis

The data from the questionnaires were arranged, coded, and archived on a computer. Statistical analysis was performed using SPSS 22.0 (IBM, Armonk, NY, USA). Regarding descriptive statistics, age, as a basic attribute, was a continuous variable that was presented as a mean and standard deviation. Categorical variables are expressed as frequencies of use and percentages. To understand the unique effect of frailty on HRQOL ($T_1$) and its predictive value, an approach was taken to control for confounding variables, which included basic attributes exhibiting a significant relationship with the outcome variable in the univariate analysis as well as HRQOL ($T_0$). Subsequently, the effect of frailty on the different domains of HRQOL was analyzed using a hierarchical regression model. The two sets of variables were entered sequentially: (1) the basic attribute variables that had a significant effect on the specific domain of HRQOL in univariate analysis and HRQOL ($T_0$) and (2) the frailty variable.

## Results

### Basic demographic attributes of elderly patients with cancer

This study included 178 elderly patients with cancer from the original database who met the inclusion criteria, with a mean age of 71.70 years (SD = 5.46 years, range = 65–96 years). Most patients were aged ≥ 70 years (n = 99, 55.6%), female (n = 96, 53.9%), and married (n = 142, 79.8%). Regarding education, most patients were elementary school graduates (n = 117, 65.7%). Most participants were unemployed (n = 155, 87.1%). Regarding primary caregivers, half the patients (89, 50%) were cared for by their spouses, whereas the other half were cared for by non-spouses. Lymphoma was the most common cancer type among patients, accounting for 38.8% (69 cases) of the cases. Concerning cancer stage, 13 patients (7.3%) had stage I cancer, 64 patients (36.0%) had stage II cancer, 66 patients (37.1%) had stage III cancer, and 33 patients (18.5%) had stage VI cancer. All patients underwent chemotherapy, and 109 patients had solid tumors and received chemotherapy after surgery. Regarding the ECOG performance status, 102 patients (57.3%) had a score of 0. Most patients did not smoke (68.0%), drink (74.2%), or chew betel nuts (89.1%). Most patients (75.3%) had other chronic diseases (Table 2). All patients received traditional chemotherapy via injection, and with the exception of one patient who received it via nasogastric tube, they could eat orally. Three patients underwent a colostomy.

**Table 2. Demographic characteristics of elderly patients with cancer (N = 178).**

| Variable | *Mean* | *SD* | *n* | % |
|---|---|---|---|---|
| Age (range: 65–96 years) | 71.70 | 5.46 | | |
| ≥ 70 years | | | 99 | 55.6 |
| 65–69 years | | | 79 | 44.4 |
| Sex | | | | |
| Male | | | 82 | 46.1 |
| Female | | | 96 | 53.9 |
| Marital status | | | | |
| Married | | | 142 | 79.8 |
| Others | | | 36 | 20.2 |
| Education | | | | |
| Elementary school or under | | | 117 | 65.7 |
| Junior high school or above | | | 61 | 34.3 |
| Occupation | | | | |
| Yes | | | 23 | 12.9 |
| No | | | 155 | 87.1 |
| Primary caregiver | | | | |
| Spouse | | | 89 | 50.0 |
| Non-spouse | | | 89 | 50.0 |
| Cancer type | | | | |
| Lymphoma | | | 69 | 38.8 |
| Breast cancer | | | 40 | 22.5 |
| Upper gastrointestinal cancer | | | 18 | 10.1 |
| Lower gastrointestinal cancer | | | 30 | 16.9 |
| Lung cancer | | | 10 | 5.6 |
| Other | | | 11 | 6.2 |
| Cancer stage | | | | |
| Stage I | | | 13 | 7.3 |
| Stage II | | | 64 | 36.0 |
| Stage III | | | 66 | 37.1 |
| Stage IV | | | 33 | 18.5 |
| Missing value | | | 2 | 1.1 |
| ECOG performance | | | | |
| 0 point | | | 102 | 57.3 |
| 1 point and above | | | 76 | 42.7 |
| Smoking | | | | |
| Yes | | | 57 | 32.0 |
| No | | | 121 | 68.0 |
| Drinking | | | | |
| Yes | | | 46 | 25.8 |
| No | | | 132 | 74.2 |
| Betel nut chewing | | | | |
| Yes | | | 18 | 10.9 |
| No | | | 160 | 89.1 |
| Chronic diseases | | | | |
| Yes | | | 134 | 75.3 |
| No | | | 44 | 24.7 |

Note. ECOG: Eastern Cooperative Oncology Group.

## Prevalence of frailty in elderly patients with cancer

This study used the CGA to assess frailty. The assessment included domains such as nutrition (MNA-SF), mood (GDS-4), functional status (ADL and IADL), comorbidity (CCI), polypharmacy, social support, and falls. The order of impairment domains was nutrition, mood, functional status, comorbidity, polypharmacy, social support, and falls (Table 3). A systematic literature review [16] found that most scholars used seven omains [26–30] or five domains [31–33] to perform CGA and failed to meet the criteria for two domains to define frailty. When seven domains were used for the CGA, 110 participants (61.8%) met the criteria for frailty. Contrastingly, when the five domains with the highest impairment rates were considered for assessment, 103 participants (57.9%) met the criteria for frailty (Table 4).

## The early predictive value of frailty assessment for patients' HRQOL

After controlling for the basic variables with a significant effect on HRQOL at baseline in step 1, the researcher included the presence or absence of frailty based on five domains of the CGA. In step 2, a regression analysis was performed on the various domains of HRQOL. The presence or absence of frailty was a significant predictor only for the illness burden domain of HRQOL.

The results indicated that 31.1–56.7% of the variance in the seven HRQOL domains could be explained by demographic variables and the presence or absence of frailty. Further, the presence or absence of frailty accounted for 1.4% of the variance in the illness burden domain ($\beta$ = 9.5%, p < .05) for post-chemotherapy HRQOL in elderly patients with cancer (Table 5) and was a significant predictor for this particular domain.

## Discussion

This study assessed frailty using five domains of the CGA with the highest impairment rate (nutrition, mood, functional status, comorbidity, and polypharmacy) and used impairment in more than two domains as the defining criteria for frailty. The results showed that 57.9% of the participants met the criteria for frailty in two domains, consistent with the results of Mohile et al. (with a prevalence rate of 60%) [32]. However, Wedding et al. also performed a CGA using five domains and used impairment in more than two domains to estimate frailty. Their study found a frailty prevalence rate of 50% [33], which is lower than that reported in this study. The researcher suggested that although the number of CGA domains and judgment criteria of the two studies were the same, this study followed the suggestion of the ASCO to

**Table 3. Impairment of various domains of comprehensive geriatric assessment (N = 178).**

| Variable | With impairment | | Without impairment | |
|---|---|---|---|---|
| | *n* | % | *n* | % |
| Nutrition (MNA-SF) | 97 | 54.5 | 81 | 45.5 |
| Mood (GDS-4) | 80 | 44.9 | 98 | 55.1 |
| Functional status (ADL & IADL) | 76 | 42.7 | 102 | 57.3 |
| Comorbidity (CCI) | 42 | 23.6 | 136 | 76.4 |
| Polypharmacy | 39 | 21.9 | 139 | 78.1 |
| Social support | 18 | 10.1 | 160 | 89.9 |
| Falls | 10 | 5.6 | 168 | 94.4 |

Note. ADL, activities of daily living; IADL, instrumental activities of daily living; MNA-SF, Mini Nutritional Assessment-Short Form; CCI, Charlson Comorbidity Index; GDS, Geriatric Depression Scale.

**Table 4. Frailty assessment results (N = 178).**

| Variable | Frail | | Non-frail | |
|---|---|---|---|---|
| | *n* | % | *n* | % |
| Frailty assessment (seven domains of the CGA) | 110 | 61.8 | 68 | 38.2 |
| Frailty assessment (five domains of the CGA) | 103 | 57.9 | 75 | 42.1 |

Note. CGA: comprehensive geriatric assessment.

consider one item of ADL or IADL to estimate the impairment of functional status [19]. However, Wedding et al. suggested that impairment of both ADL and IADL can be regarded as impairment of functional status, resulting in a lower rate of frailty [33].

If frailty was assessed using seven domains (nutrition, mood, functional status, comorbidity, polypharmacy, social support, and falls) and impairment in more than two domains was used as the judgment criterion for frailty, 61.8% of the participants were determined to be frail, similar to Baitar et al. [26] (64%). However, Owusu et al. also used seven domains and impairment in more than two domains as judgment criteria for frailty and discovered that the prevalence rate of frailty was only 43% [27], which is lower than that in this study. This may be because the cancer stages of the participants were more severe. Ninety-nine elderly patients with cancer (55.6%) were classified as either stage III or stage IV. Nevertheless, only 47 elderly patients with cancer (41%) had stage III or stage IV cancer in Owusu et al. [27]. Moreover, Valéro et al. [30] used seven domains of the CGA, but employed the impairment of more than four domains as a judgment criterion for frailty. The results showed that the prevalence of frailty was only 10% [30], which was significantly different from that in this study. Inconsistencies in the defining criteria for frailty affect its prevalence. Unfortunately, the SIOG and NCCN have not offered specific suggestions on the defining criteria of the CGA. This study only followed the suggestions of most scholars to use the impairment of more than two CGA domains to judge frailty [16, 26, 27]. The frailty assessment results using five and seven domains of the CGA were similar (57.9% vs. 61.8%) in this study, and the medical environment in Taiwan is characterized by a high level of busyness. Further, the CGA requires a considerable amount of time and resources to be administered by adequately trained assessors [22]. Therefore, this study suggests the adoption of the five domains of the CGA as a more practical and advantageous approach for frailty assessment by clinical medical and nursing personnel. If a large-scale, multicenter, or multinational study is to be conducted in the future, it will be necessary to reach a consensus on the number of CGA domains and define the criteria before a comparison can be performed.

Experts believe that the CGA should include multiple domains [19–21] but suggest that there is no need to assess all domains [23]. A systematic review [16] found that most scholars used seven [26–30] and five domains [31–33] of the CGA to assess frailty. This review summarized relevant studies on CGA using seven domains and found that the following five domains had the highest impairment rates: functional status [6, 26, 27, 29], nutrition [6, 26, 29], mood [6, 26, 27], comorbidity [6, 26, 27], and polypharmacy [6]. These domains are consistent with the five domains with the highest impairment rates in this study (nutrition, mood, functional status, comorbidity, and polypharmacy), indicating that the important dimensions of frailty could be similar between Western and Eastern populations.

According to the results of the regression analysis, the presence or absence of frailty only explained 0.1–1.4% of the variance in various domains of post-chemotherapy HRQOL in elderly patients with cancer. There were no significant differences in most domains (p > .05), except for the illness burden domain (p < .05). The results showed that the presence or

**Table 5. Hierarchical regression analysis on patients' health-related quality of life (N = 178).**

| Predictor variables | Illness burden domain | | | | Future worries domain | | | | Mobility domain | | | | Worries about others domain | | | |
|---|---|---|---|---|---|---|---|---|---|---|---|---|---|---|---|---|
| | Step 1 | | Step 2 | | Step 1 | | Step 2 | | Step 1 | | Step 2 | | Step 1 | | Step 2 | |
| | β | t | β | t | β | t | β | t | β | t | β | t | β | t | β | t |
| Constant | -9.55 | -1.06 | -7.95 | -.89 | -.46 | -.05 | -.70 | -.07 | 9.66 | 2.00* | 9.50 | 1.97 | -16.11 | -1.63 | -16.54 | -1.66 |
| Sex | -.38 | -.09 | -.64 | -.16 | 2.32 | .73 | 2.35 | .73 | | | | | 3.29 | .78 | 3.38 | .80 |
| Primary caregiver | 7.71 | 1.95 | 6.85 | 1.75 | 2.13 | .60 | 2.53 | .64 | | | | | -1.52 | -.33 | -1.34 | -.29 |
| ECOG | 5.74 | 1.50 | 2.74 | .69 | 1.06 | .34 | 1.44 | .44 | -2.57 | -.79 | -3.64 | -1.10 | 7.07 | 1.75 | 7.92 | 1.87 |
| Marital status | | | | | 3.70 | .88 | 3.70 | .88 | | | | | 13.37 | 2.41* | 13.52 | 2.43* |
| Education | | | | | -4.80 | -1.53 | -4.77 | -1.52 | | | | | | | | |
| Chronic disease | | | | | .41 | .08 | -.47 | -.09 | 2.75 | .84 | 1.70 | .51 | | | | |
| HRQOL(T₀) | .70 | 13.27*** | .68 | 12.78*** | .74 | 13.60*** | .74 | 13.13*** | .84 | 12.42*** | .82 | 11.88*** | .59 | 10.14*** | .59 | 10.07*** |
| Presence/absence of frailty | | | 9.50 | 2.36* | | | -1.33 | -.40 | | | 5.14 | 1.65 | | | -2.78 | -.65 |
| R² | .542 | | .557 | | .566 | | .567 | | .545 | | .552 | | .438 | | .439 | |
| F | 51.28*** | | 43.22*** | | 37.18*** | | 31.74*** | | 69.60*** | | 53.39*** | | 26.79*** | | 22.32*** | |
| Change in R² | | | .014* | | | | .001 | | | | .007 | | | | .001 | |

| Predictor variables | Maintaining purpose domain | | | | Joint stiffness | | | | Family support | | | |
|---|---|---|---|---|---|---|---|---|---|---|---|---|
| | Step 1 | | Step 2 | | Step 1 | | Step 2 | | Step 1 | | Step 2 | |
| | β | t | β | t | β | t | β | t | β | t | β | t |
| Constant | 55.12 | 4.66*** | 55.61 | 4.68*** | -3.72 | -.35 | -3.77 | -.36 | 48.23 | 3.88*** | 46.67 | 3.70*** |
| Gender | | | | | 4.65 | 1.21 | 4.67 | 1.22 | 9.37 | 1.91 | 8.88 | 1.79 |
| Primary caregiver | | | | | -.71 | -.23 | -.59 | -.19 | | | | |
| ECOG | -6.70 | -1.74 | -5.99 | -1.49 | 7.54 | 2.33* | 7.96 | 2.35* | | | | |
| Marital status | -6.56 | -1.47 | -6.37 | -1.42 | | | | | | | | |
| Education | 2.53 | .66 | 2.59 | .67 | -4.71 | -1.50 | -4.70 | -1.49 | | | | |
| Smoking | | | | | -2.37 | -.63 | -2.45 | -.65 | | | | |
| Cancer stage (ref Stage I) | | | | | | | | | | | | |
| Stage II | -4.73 | -.69 | -4.72 | -.69 | | | | | | | | |
| Stage III | -3.16 | -.46 | -3.09 | -.45 | | | | | | | | |
| Stage IV | -10.65 | -1.42 | -10.55 | -1.40 | | | | | | | | |
| Cancer type (ref Lung cancer) | | | | | | | | | | | | |
| Lymphoma | | | | | | | | | -22.84 | -2.41* | -23.09 | -2.43* |
| Breast cancer | | | | | | | | | -19.53 | -1.95 | -18.92 | -1.88 |
| Lower GI cancer | | | | | | | | | -12.86 | -1.25 | -12.98 | -1.26 |
| Upper GI cancer | | | | | | | | | -18.89 | -1.70 | -18.70 | -1.68 |
| Others | | | | | | | | | -14.37 | -1.15 | -14.43 | -1.16 |
| HRQOL(T0) | .54 | 8.52*** | .53 | 8.31*** | .55 | 9.20*** | .55 | 9.18*** | .46 | 7.00*** | .46 | 7.03*** |
| Presence/absence of frailty | | | -2.46 | -.64 | | | -1.36 | -.43 | | | 3.56 | .82 |
| R² | .420 | | .421 | | .457 | | .458 | | .308 | | .311 | |
| F | 17.57*** | | 15.37*** | | 24.05*** | | 20.54*** | | 10.82*** | | 9.53** | |
| Change in R² | | | .001 | | | | .001 | | | | .003 | |

Note.

***$p < .001$

**$p < .01$

*$p < .05$; Step 1: controlling for the basic variables with a significant effect on HRQOL, Step 2: the presence or absence of frailty (using 5 domains of CGA).

absence of frailty was an important predictor of the illness burden domain of the post-chemotherapy HRQOL in elderly patients with cancer. Most scholars have assessed frailty after chemotherapy to predict the HRQOL and found that frailty is an important predictor of HRQOL [24, 34–36]. Most researchers have used the EORTC QLQ-C30 and Functional Assessment of Cancer Therapy-General Scale (FACT-G) to measure HRQOL. Despite being developed by the European Organization for Research and Treatment of Cancer (EORTC) to evaluate the HRQOL of elderly patients with cancer, the QOL measurement tool (EORTC QLQ-ELD14) has been rarely used. Further, this study is the first to predict the HRQOL after chemotherapy (using the EORTC QLQ-ELD14) by evaluating frailty before chemotherapy in elderly patients with cancer. This makes it difficult to compare and discuss these findings with those of previous studies. It is possible that some variables not measured in this study, such as financial

toxicity, treatment side effects, and spiritual well-being, could be related to HRQOL, especially in elderly patients with cancer in advanced stages. Moreover, the scoring method of the EORTC QLQ-ELD14 differs from that of most HRQOL scales, which use the total score to assess the HRQOL of patients with cancer, such as the FACT-G [37–41], Functional Living Index-Cancer [42], QOL-Cancer Survivors [43], QOL—AntiCancer Drugs [44], and Multidimensional QOL Scale-Cancer [45]. The EORTC QLQ-ELD14 scale used in this study does not use the total HRQOL score for assessment, but utilizes the scores of various subscales to perform respective analyses [10]. This approach could have overlooked the wholeness of HRQOL [46]. The HRQOL of elderly patients with cancer is unique [3], and the EORTC QLQ-ELD14 scale has been developed accordingly [3, 10]. However, this scale has rarely been used. The FACT-G, developed by Cella et al., could be a better HRQOL assessment tool [38]. It not only has good reliability and validity [37, 39, 40], it has also been widely used by scholars in various countries around the world [37, 40, 41]. The Chinese-translated version and reliability and validity data are also available [40]. Therefore, it could be a better choice for measuring HRQOL in elderly patients with cancer and future studies should consider using this scale.

## Conclusion

This study is the first to investigate the early predictive value of frailty for HRQOL in elderly patients with cancer in Taiwan. The findings showed that the presence or absence of frailty is an important predictor of domains such as illness burden in elderly patients with cancer. Frailty was a common issue among elderly patients with cancer. Thus, it is recommended that elderly patients with cancer undergo frailty assessment before treatment. These findings could serve as a reference for treatment-related decision-making (providing anticancer treatment, reducing anticancer treatment intensity, or not providing anticancer treatment) or the development of treatment plans.

## Supporting information

**S1 Data.**
(SAV)

## Acknowledgments

The authors are grateful for the assistance provided by the Chang Gung Memorial Hospital Cancer Center for data collection. Special thanks to all the patients with cancer who participated in this study.

## Author Contributions

**Conceptualization:** Yi-Cheng Hu, Shih-Ying Chen, Wen-Chi Chou, Jen-Shi Chen, Li-Chueh Weng, Pei-Kwei Tsay, Woung-Ru Tang.

**Data curation:** Shih-Ying Chen.

**Formal analysis:** Yi-Cheng Hu, Pei-Kwei Tsay, Woung-Ru Tang.

**Investigation:** Yi-Cheng Hu, Shih-Ying Chen, Woung-Ru Tang.

**Project administration:** Wen-Chi Chou, Jen-Shi Chen.

**Validation:** Shih-Ying Chen.

**Writing – original draft:** Yi-Cheng Hu, Woung-Ru Tang.

**Writing – review & editing:** Yi-Cheng Hu, Shih-Ying Chen, Wen-Chi Chou, Jen-Shi Chen, Li-Chueh Weng, Pei-Kwei Tsay, Woung-Ru Tang.

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
