## [Decision Letter · Decision Letter 0]

6 Mar 2023

PONE-D-22-31278Frailty as an Early Predictive Value for Health-Related Quality of Life among Elderly Cancer Patients Receiving Curative ChemotherapyPLOS ONE

Dear Dr. Tang,

Thank you for submitting your manuscript to PLOS ONE. After careful consideration, we feel that it has merit, but there is space to imrove your work to fully meet PLOS ONE’s publication criteria as it currently stands. Therefore, we invite you to submit a revised version of the manuscript that addresses the points raised during the review process.

Please submit your revised manuscript within 2 weeks. If you will need more time than this to complete your revisions, please reply to this message or contact the journal office at plosone@plos.org. Please include the following items when submitting your revised manuscript:A rebuttal letter that responds to each point raised by the academic editor and reviewer(s). You should upload this letter as a separate file labeled 'Response to Reviewers'.A marked-up copy of your manuscript that highlights changes made to the original version. You should upload this as a separate file labeled 'Revised Manuscript with Track Changes'.An unmarked version of your revised paper without tracked changes. You should upload this as a separate file labeled 'Manuscript'.

We look forward to receiving your revised manuscript.

Kind regards,

Chong-Chi Chiu

Academic Editor

PLOS ONE

Journal Requirements:

**

**

Additional Editor Comments (if provided):

(1) Reviewer 1 pointed out the limitations of this study. Please revise the manuscript as suggested.

(2) Health-Related Quality of Life could be affected by many factors, e.g., the condition of nutrition support (oral or tube enteral feeding), the need for stoma for stool passage, and the methods of chemotherapy (oral or intravenous route). The lack of complete data would lead to a bias in prediction results via this pre-chemotherapy frailty assessment. Please provide more information.

Reviewers' comments:

Reviewer's Responses to Questions

**Comments to the Author**

1. Is the manuscript technically sound, and do the data support the conclusions?

Reviewer #1: Partly

Reviewer #2: Yes

2. Has the statistical analysis been performed appropriately and rigorously? 

Reviewer #1: No

Reviewer #2: Yes

3. Have the authors made all data underlying the findings in their manuscript fully available?

Reviewer #1: No

Reviewer #2: Yes

4. Is the manuscript presented in an intelligible fashion and written in standard English?

Reviewer #1: Yes

Reviewer #2: Yes

5. Review Comments to the Author

Reviewer #1: This study aimed to investigate the early predictive value of pre-chemotherapy frailty assessment for post-chemotherapy health-related quality of life among elderly cancer patients receiving curative chemotherapy.

However, the study has several limitations that compromise its validity. The sample size is relativly small, while unknown cancer types are grouped together (we only know that 38% of the patients are reported to have lymphoma). There is no information on surgery and chemotherapy completeness and duration. The tested associations were not adjusted for important variables, like cancer type and stage (or it is not reported). Those factors can explain part of the variance in the post-chemotherapy quality of life, thus making present inferences not reliable. The methods and results are unclear and poorly reported.

Specific comments:

The authors should include a reference to the study from which the data was obtained as described in subsection 2.1 of the methods section.

In section 2.4, the authors should clearly define what the "basic attributes" are and how they were selected for the final model.

In Table 2, the numbers for education do not add up to 178. The authors should explain if there were missing data and how they used this variable in the linear regression model. The same issue applies to the cancer stage variable.

In section 3.2, rows 222-224, the meaning of the ">" sign in front of the frailty characteristics is not clear. The authors should avoid double reporting of the same numbers in the text and tables.

In section 3.2, rows 227-230, it is unclear which frailty criteria were eventually used, five or seven domains?

In section 3.3 and Table 5, the authors used paired t-tests to determine the significance of differences in HRQOL domains before and after chemotherapy. However, the distribution of the scores and their normality were not evaluated, which raises questions about the appropriateness of using paired t-tests.

In section 3.3, the authors reported significant differences in 'mobility' and 'illness burden' QoL before and after chemotherapy, but not in 'future worries'. In section 3.4 and Table 6, linear regression was used to test the effects of demographic characteristics and frailty status on post-chemotherapy QoL for the domains of 'mobility', 'future worries', and 'illness burden'. The reasoning for this selection of domains is not clear.

In section 3.4, the concepts of "hierarchies" 1 and 2 should be clearly explained. Do they correspond to steps 1 and 2 from Table 6? It is also unclear if the associations were adjusted for baseline QoL. Furthermore, the selection of variables for the linear regression model is unclear. As it is reported in table 2, the authors collected more information about patients, than they included into models. Not clear why.

In Table 6, the steps 1 and 2 are not clearly specified. The authors should explain this clearly in the legend. The association is not adjusted for age, cancer type, and stage, which could have explained a part of the variance.

The discussion should focus on the findings of this study in relation to its aim, which is "the predictive value of pre-chemotherapy frailty assessment for post-chemotherapy health-related quality of life", rather than on comparing the descriptive characteristics of the cohort with other studies.

The statement "There was no statistically significant difference in most of the domains (p > .05), except for the mobility, future worries, and illness burden domains (p < .05)" in rows 385-387 contradicts the results reported in the results section "However, there were no statistically significant differences in other domains of HRQOL (worries about others, future worries, maintaining purpose, joint stiffness, and family support) (p >.05) (Table 5)". 'Future worries' was mentioned in both, which is inconsistent.

Reviewer #2: Thankyou for the opportunity to review your paper

I think it is well structured and asks a unique question. Thus i think it is a useful addition to the literature

I would recommend acceptance without modification

6. PLOS authors have the option to publish the peer review history of their article (what does this mean?). If published, this will include your full peer review and any attached files.

Reviewer #1: No

Reviewer #2: **Yes: **Simon Richards

---

## [Author Response · Author response to Decision Letter 0]

20 Apr 2023

Dear editor and reviewers, 

We truly appreciate the opportunity to revise our manuscript titled, “Frailty as an Early Predictive Value for Health-Related Quality of Life among Elderly Cancer Patients Receiving Curative Chemotherapy.” The constructive and valuable comments from the reviewers have provided guidance, which we believe has improved the quality of our manuscript. The manuscript has been revised to address all the reviewers’ comments and concerns, including revisions for grammar and wording as well as the addition of references in response to the reviewers’ questions.

To facilitate the review, all changes to the manuscript are in red bold type. We firmly believe that the publication of this paper will make a significant contribution to the literature on the predictive value of frailty on the quality of life of elderly patients with advanced stages cancer. We hope that the manuscript is now acceptable for publication in PLOS ONE. Thank you very much for your time and attention.

Sincerely, 

Woung-Ru Tang

Professor, School of Nursing, 

Chang Gung University, Taoyuan, Taiwan, ROC.

---

## [Editor Report · Decision Letter 1]

28 Apr 2023

PONE-D-22-31278R1Frailty as an Early Predictive Value for Health-Related Quality of Life among Elderly Cancer Patients Receiving Curative ChemotherapyPLOS ONE

Dear Dr. Tang,

Thank you for submitting your manuscript to PLOS ONE. After careful consideration, we feel that it has improvement after 1st revision. Therefore, we invite you to submit a revised version of the manuscript that addresses the points raised during the review process (MINOR REVISION).

We look forward to receiving your revised manuscript.

Kind regards,

Chong-Chi Chiu

Academic Editor

PLOS ONE

Journal Requirements:

Additional Editor Comments:

1. The authors have revised the text and tables based on the suggestions.

2. The similarity rate is 21%, which needs revision.

3. Please provide proof of English editing by a native English or a professional institution.
---

## [Author Response · Author response to Decision Letter 1]

6 May 2023

Dear Editor, 

We truly appreciate the opportunity to revise our manuscript again, titled “The Early Predictive Value of Frailty for Health-related Quality of Life Among Elderly Patients with Cancer Receiving Curative Chemotherapy.” The constructive and valuable comments from the editor have provided guidance, which we believe has improved the quality of our manuscript. The manuscript has been revised to address the editor’s comments and concerns, including revisions for grammar and wording as well as checking the references in response to the editor’s questions.

To facilitate the review, all changes to the manuscript are in red bold type. We firmly believe that the publication of this paper will make a significant contribution to the literature on the predictive value of frailty on the quality of life of elderly patients with advanced stages cancer. We hope that the manuscript is now acceptable for publication in PLOS ONE. Thank you very much for your time and attention.

Sincerely, 

Woung-Ru Tang

Professor, School of Nursing, 

Chang Gung University, Taoyuan, Taiwan, ROC.

---

## [Editor Report · Decision Letter 2]

4 Jun 2023

The Early Predictive Value of Frailty for Health-related Quality of Life Among Elderly Patients with Cancer Receiving Curative Chemotherapy

PONE-D-22-31278R2

Dear Correspondent Tang,

We’re pleased to inform you that your manuscript has been judged scientifically suitable for publication and will be formally accepted for publication once it meets all outstanding technical requirements.

Kind regards,

Chong-Chi Chiu

Academic Editor

PLOS ONE

---

## [Editor Report · Acceptance letter]

24 Jul 2023

PONE-D-22-31278R2 

The Early Predictive Value of Frailty for Health-related Quality of Life Among Elderly Patients with Cancer Receiving Curative Chemotherapy 

Dear Dr. Tang:

I'm pleased to inform you that your manuscript has been deemed suitable for publication in PLOS ONE. Congratulations! Your manuscript is now with our production department. 

Kind regards, 

on behalf of

Professor Chong-Chi Chiu 

Academic Editor

PLOS ONE